# An Experimental Study of Microchannel and Micro-Pin-Fin Based On-Chip Cooling Systems with Silicon-to-Silicon Direct Bonding

**DOI:** 10.3390/s20195533

**Published:** 2020-09-27

**Authors:** Yunlong Qiu, Wenjie Hu, Changju Wu, Weifang Chen

**Affiliations:** School of Aeronautics and Astronautics, Zhejiang University, Zheda Road, Hangzhou 310027, China; qyl1992@zju.edu.cn (Y.Q.); hwj1997@zju.edu.cn (W.H.); chenwfnudt@163.com (W.C.)

**Keywords:** power IC, electronic cooling, microchannel, micro pin fin, thermal resistance, MEMS

## Abstract

This paper describes an experimental study of the cooling capabilities of microchannel and micro-pin-fin based on-chip cooling systems. The on-chip cooling systems integrated with a micro heat sink, simulated power IC (integrated circuit) and temperature sensors are fabricated by micromachining and silicon-to-silicon direct bonding. Three micro heat sink structures: a microchannel heat sink (MCHS), an inline micro-pin-fin heat sink (I-MPFHS) and a staggered micro-pin-fin heat sink (S-MPFHS) are tested in the Reynolds number range of 79.2 to 882.3. The results show that S-MPFHS is preferred if the water pump can provide enough pressure drop. However, S-MPFHS has the worst performance when the rated pressure drop of the pump is lower than 1.5 kPa because the endwall effect under a low Reynolds number suppresses the disturbance generated by the staggered micro pin fins but S-MPFHS is still preferred when the rated pressure drop of the pump is in the range of 1.5 to 20 kPa. When the rated pressure drop of the pump is higher than 20 kPa, I-MPFHS will be the best choice because of high heat transfer enhancement and low pressure drop price brought by the unsteady vortex street.

## 1. Introduction

In recent years, developments in new fabricating and packaging technologies, such as extreme ultra-violet (EUV) lithography and three-dimensional integrated circuits (3D ICs), have made possible the continuation of Moore’s Law [1,2,3,4]. However, while it improves performance, the increased transistor density of microchips increases power density, which is a cause of microchip thermal management becoming one of the bottlenecks in the further development of semiconductor technology [5,6]. The next generation of microchips is expected to produce heat fluxes higher than 150 W/cm^2^ with localized hot spots of up to 1000 W/cm^2^ [7,8]. For such high heat fluxes, traditional air-cooling or separated liquid-cooling technology is no longer suitable because of the large thermal resistance from microchip to working fluid.

On-chip liquid-cooling technology is a high-performance cooling technology, which etches microchannels directly on the backside of the microchip and simplifies the multistep thermal-conduction process (chip-thermal interface material-package shell- thermal interface material-heat sink) to a single-step thermal-conduction process (chip-heat sink) [9,10,11]. As the thermal-conduction steps are greatly reduced, thermal-conduction resistance becomes much lower than thermal-convection resistance, which means that the cooling capability of the on-chip liquid-cooling system is dominated by the convection heat transfer performance between the microchannel heat sink and the working fluid. Thus, the optimization of the microchannel structure is key to improving the cooling capability of the on-chip liquid-cooling system.

The most simple and effective way to do this is to reduce the hydraulic diameter of the microchannels [12,13]. However, reducing the hydraulic diameter, while improving heat transfer, significantly increases the pressure drop, which may cause leakage of the liquid-cooling system and destroy the electronic equipment. In addition, existing micro pumps, which are ideal driving elements in micro cooling systems because of their small size and good integrability, are unlikely to provide such a high pressure drop [14]. Therefore, a major focus of studies on microchannel heat sinks is to improve heat transfer with a reasonable pressure drop through geometric improvements.

The optimization of the microchannel shape has been widely studied and the research literatures show that optimizing the cross-sectional shape of microchannels can bring a certain heat transfer enhancement [15,16,17,18]. However, the thermal boundary layer of smooth and straight microchannels rapidly become thicker with increasing flow distance, which significantly limits the heat transfer enhancement offered by cross-section optimization [19,20]. Inspired by the concept of redeveloping the thermal boundary layer, researchers have proposed the idea of a micro-pin-fin heat sink that uses short micro pin fins instead of long and straight microchannels to interrupt the thermal boundary layer and improve flow disturbance [21,22]. Both experimental and numerical studies on circular micro-pin-fin heat sinks show that staggered micro-pin-fin heat sinks have a much higher Nusselt number than inline micro-pin-fin heat sinks and straight rectangular microchannel heat sinks but that the tortuous flow path in the staggered micro-pin-fin array also results in a high pressure drop price [23,24,25,26,27,28,29]. In order to compare the performance of microchannel and micro-pin-fin heat sinks, some researchers have used a thermal performance index (TPI) to comprehensively evaluate the Nusselt number and pressure drop changes caused by geometric improvements [26,27]. However, the thermal performance index is derived for a fully developed flow with a constant pumping power for which the product of the friction factor and Reynolds number is constant. This assumption is inapplicable to micro heat sinks because of the non-negligible entrance effect. Additionally, most reports in the literature concerns themselves more with heat transfer performance per convection area rather than with the temperature rise of the simulated power IC and most conclusions about the performance of microchannel and micro-pin-fin heat sinks in such literature are based on three-sided heating cases, which use glass as cover plates [23,24,25,26]. Whether these conclusions are applicable to four-sided heating cases has not been systematically studied.

Thus, in this study, on-chip cooling systems fabricated through micromachining and silicon-to-silicon bonding are used to study the cooling capabilities of four-sided heated micro heat sinks. Three heat sink structures are tested in the Reynolds number range of 79.2 to 882.3, namely, a microchannel heat sink (MCHS), an inline micro-pin-fin heat sink (I-MPFHS) and a staggered micro-pin-fin heat sink (S-MPFHS). The application situations of these three heat sinks are summarized according to the experimental results, which provides theoretical guidance for the design of on-chip liquid-cooling systems.

## 2. Materials and Methods

### 2.1. Experimental Setup

As shown in Figure 1, the experimental system is composed of a thermostatic water tank, a filter, a gear pump, a flow meter, a test chip and a data acquisition system. The temperature of the thermostatic water tank is kept at 298.15 K with a temperature fluctuation of less than 0.05 K. A water filter with a filtration precision of 15 μm is installed before the gear pump to remove impurities. A temperature port and a pressure port are installed at both the inlet and the outlet of the test chip to measure the temperature rise and pressure drop of the working fluid (deionized water). The pressure ports are closer to the test chip than the temperature ports to avoid an additional pressure drop caused by the installation of thermocouples.

### 2.2. Test Chip

The test chips have four layers, including a silicon substrate (200 μm), a microchannel or micro-pin-fin layer (300 μm), a silicon cover (500 μm) and a Ti (20 nm)/Pt (200 nm) electrode layer. Figure 2 shows structural illustrations of the test chips. Three types of heat sinks are investigated in this study, namely, a microchannel heat sink (MCHS), an inline micro-pin-fin heat sink (I-MPFHS) and a staggered micro-pin-fin heat sink (S-MPFHS).

As shown in Figure 2a, the microchannel array is 6.6 mm wide and 14.4 mm long. It includes eleven microchannels with a width and interval of 300 μm. As shown in Figure 2b,c, the inline and the staggered micro-pin-fin arrays have the same size as the microchannel array. It has twenty-four rows and all the micro pin fins are 300 μm in diameter. The transverse and longitudinal distances between the centers of adjacent pin fins are 600 μm. As shown in Figure 2d, the Ti/Pt electrode layer is deposited on the backside of the substrate. It consists of thirteen thin-film resistors, including six heaters (2.9 mm × 4.6 mm) and seven sensors (300 μm × 345 μm). The heaters cover the entire microchannel or micro-pin-fin array and act as a simulated power IC (integrated circuit). They are also used to measure the average temperature of the heat sink. Each heater provides a heat flux of 100 W/cm^2^. The sensors are placed along the centerline of the test chips to measure the temperature distribution of the power IC along the flow direction. The distance between the centers of adjacent sensors is 2.16 mm.

The fabrication process of the test chips is shown in Figure 3 and is described as follows:
(a)Step 1 ~ Step 2: a thick photoresist film is span-coated on the silicon substrate and then patterned by photolithography.(b)Step 3 ~ Step 4: the silicon substrate is etched with the DRIE (Deep Reactive Ion Etching) to create the microchannel or micro-pin-fin layer and then the photoresist is removed from the silicon.(c)Step 5: the inlet and the outlet on the silicon cover are drilled by laser etching.(d)Step 6: silicon-to-silicon direct bonding is achieved using high temperature melting.(e)Step 7 ~ Step 8: a 50 nm-thick SiO_2_ layer is grown on both sides of the chip by thermal oxidation to serve as electrical insulation and a 500 nm-thick Si_x_N_y_ is deposited on the backside of the chip by PECVD (Plasma Enhanced Chemical Vapor Deposition) to improve electrical insulativity.(f)Step 9 ~ Step 13: a 20 nm/200 nm-thick Ti/Pt electrode layer is sputtered on the backside of the silicon substrate using magnetron sputtering and then patterned by photolithography and IBE (Ion Beam Etching).(g)Step 14: the fabricated chip is bonded onto a customized printed circuit board (PCB) using Flip-chip technology.

Figure 4 shows the morphology and microstructure of the fabricated microchannel and micro pin fin with scanning electron microscopy (SEM). The pictures show that the microchannel and micro-pin-fin surfaces are smooth and the perpendicularity is high, which minimizes the influence of fabrication error on the experimental results.

### 2.3. Data Reduction

The Reynolds number of the incoming flow is defined as:(1)Re=ρU¯Lμ=ρQLμAmin,
where *Q* is the flow rate, *L* is the characterized length (the hydraulic diameter for the microchannel heat sink and the fin diameter for the micro pin fin heat sink, both equal to 300 μm), *A_min_* is the minimum vertical cross-sectional area of the flow region (0.99 mm^2^ for both the microchannel and the micro-pin-fin heat sinks).

The average heat transfer coefficient is defined as:(2)$$h=\frac{q}{A_{c}(\bar{T_{h}}-T_{f})},$$
where *q* is the effective heating power, which is calculated by energy conservation:(3)$$q==\rho c_{p}Q(T_{o}-T_{i}),$$
where *T_i_* and *T_o_* are the average temperatures of the fluid at the chip inlet and outlet, respectively. *A_c_* is the effective convection area of the heat sink. (190.08 mm^2^ for the microchannel heat sink, 231.72 mm^2^ for the inline micro-pin-fin heat sink and 233.88 mm^2^ for the staggered micro-pin-fin heat sink). $\bar{T_{h}}$ is the average temperature of the heat sink and is measured by the heaters. *T_f_* is the average temperature of the working fluid and is calculated by:(4)$$T_{f}=\frac{T_{i}+T_{o}}{2}.$$

The average Nusselt number is defined as:(5)$$\mathrm{Nu}=\frac{hL}{k_{f}}=\frac{\rho c_{p}QL(T_{o}-T_{i})}{A_{c}k_{f}(\bar{T_{h}}-T_{f})}.$$

The average thermal resistance is defined as:(6)$$\theta =\frac{\bar{T_{h}}-T_{i}}{q}=\frac{\bar{T_{h}}-T_{i}}{\rho c_{p}Q(T_{o}-T_{i})}.$$

The resistance-temperature relations of the temperature sensors and the heaters are calibrated by a thermostatic water tank and a digital multimeter. The temperature accuracy of the water tank is ±0.15 K. The resistance measuring error of the digital multimeter is ±0.02 Ω. The four-wire resistance measurement is used to eliminate the influence of wire resistance. The fitted resistance-temperature relations of the sensors and heaters are given in Equations (7) and (8), respectively:

Sensors:(7)$$R_{s}=\alpha_{s}T_{s}+\beta_{s},\;\alpha_{s}=0.3505\;\Omega /K,\;\beta_{s}=56.57\;\Omega .$$

Heaters:(8)$$R_{h}=\alpha_{h}T_{h}+\beta_{h},\;\alpha_{h}=0.2766\;\Omega /K,\;\beta_{h}=44.15\;\Omega .$$

Figure 5 compares the fitted calibration curves with the measurements of resistance. The results show that the maximum linearity errors of the calibration curves are ±0.21 K for sensors and ±0.23 K for heaters. Thus, according to the error accumulation principle, the accuracy of the temperature sensors is ±0.42 K and that of the heaters is ±0.45 K.

The experimental uncertainties of the derived parameters are calculated using the method proposed by Moffat [30]. The calculation formula is:(9)$$\mathrm{For}\;y=f(x_{1},x_{2},\ldots ,x_{n}),\frac{\Delta y}{y}=\sqrt{{\left(\frac{\partial y}{\partial x_{1}}\cdot \frac{\Delta x_{1}}{y}\right)}^{2}+{\left(\frac{\partial y}{\partial x_{2}}\cdot \frac{\Delta x_{2}}{y}\right)}^{2}+\cdot \cdot \cdot +{\left(\frac{\partial y}{\partial x_{n}}\cdot \frac{\Delta x_{n}}{y}\right)}^{2}}$$

The results are shown in Table 1.

## 3. Results and Discussions

Figure 6 shows the variations of the average Nusselt number (Nu) with the Reynolds number (Re). The results show that Nu of the staggered micro-pin-fin heat sink (S-MPFHS) is much higher than those of the microchannel heat sink (MCHS) and the inline micro-pin-fin heat sink (I-MPFHS). In S-MPFHS, the micro pin fins in the back row are in the middle of the two adjacent micro pin fins in the front row and so the fluid flowing through the front-row micro pin fins will impinge the back-row micro pin fins. According to the field synergy principle proposed by Guo [31], the impingement flow greatly decreases the angle between the velocity vector and temperature gradient, which significantly enhances the heat transfer performance of the heat sink. However, it is worth noting that the heat transfer advantage of S-MPFHS over MCHS and I-MPFHS gradually decreases with decreasing Re for Re < 300. The reason for this phenomenon is that when Re is low, the disturbance generated by the micro pin fins is strongly weakened by the thick boundary layers of the upper and lower endwalls of the micro-pin-fin layer. As Re increases, the boundary layers of the upper and lower endwalls become thinner and the impact of the endwall effect gradually diminishes, which brings on a rapid increase of heat transfer performance. In addition, by comparing the Nu–Re curves of MCHS and I-MPFHS in Figure 6, it can be seen that Nu of I-MPFHS is slightly lower than that of MCHS when Re is lower than 550. This is mainly because the flow structure in the separation zones of the inline micro pin fins is a steady large-scale vortex pair at a low Reynolds number. The rotation speed of the vortex pair is slow and the momentum exchange between the separation zones and the main flow is weak, which results in poor heat transfer performance. However, when Re is higher than 550, the flow stability in the separation zones of the inline micro pin fins decreases and the flow structure in the separation zones gradually evolves from a steady vortex pair to an unsteady Kármán vortex street. Similar phenomena are observed in the experimental works of Jung [32] and Wang [33]. Affected by periodic vortex generation and shedding of the Kármán vortex street, the momentum and energy exchanges between the separation zones and the main flow are greatly enhanced. As a result, Nu of I-MPFHS increases rapidly with increasing Re and exceeds that of MCHS. Under the condition of Re = 814.4, Nu of I-MPFHS reaches 11.70, while that of MCHS is only 8.54.

According to the definition of Nu in Equation (6), it can be seen that the effects of fluid temperature rise along the flow direction and the convection area differences between MCHS, I-MPFHS and S-MPFHS are excluded in calculating Nu. Because of this, Nu represents the average cooling capacity of the fluid-solid interface of the microchannel/micro-pin-fin array rather than the actual cooling capacity of a heat sink. As defined in Equation (7), the average thermal resistance (*θ*) represents the average temperature rise of the simulated power IC per unit power input. Its calculation takes into account the effects of both fluid temperature rise and convection area difference, so that *θ* is more reflective of the actual cooling capability of a heat sink than Nu. Figure 7 shows variations of *θ* with Re. As the effective convection area of MCHS (190.08 mm^2^) is smaller than that of I-MPFHS (231.72 mm^2^), it can be seen that the actual cooling capabilities of MCHS are lower than those of I-MPFHS, even at low Re. For S-MPFHS, due to its high average Nusselt number and large effective convection area (233.88 mm^2^), the actual cooling capacity of S-MPFHS is significantly higher than that of MCHS and I-MPFHS at the same Re, especially at moderate Re (271.5 < Re < 542.9).

Figure 8 compares the temperature distributions along the centerline of the simulated power IC under the conditions of Re = 339.3 and Re = 678.7. The results show that the temperature uniformity of S-MPFHS is better than those of MCHS and I-MPFHS. In S-MPFHS, the boundary layers on the endwalls and the surface of each micro pin fin are always interrupted and re-developed, so the local heat transfer performance difference between the upstream area and the downstream area is small, which means that temperature rise along the centerline of the simulated power IC is caused mainly by the fluid temperature rise along the flow direction. However, in MCHS, the boundary layers of the straight rectangular microchannels are less affected. They grow thicker rapidly as the flow distance increases, which results in a large local heat transfer performance difference between the upstream area and the downstream area of MCHS. In addition, the maximum temperature difference of MCHS is about twice the value of S-MPFHS at Re = 339.3, while it is about triple at Re = 678.7. This demonstrates that the improvement of temperature uniformity induced by increasing Re is more significant for S-MPFHS than for MCHS. As for I-MPFHS, when Re is lower than 550, the separation zones in the back of the micro pin fins keep steady and they participate only minimally in the heat transfer process. Thus, the heat transfer performance of I-MPFHS is similar to that of MCHS. As shown in Figure 8a, the temperature distribution of I-MPFHS at Re =339.3 is close to that of MCHS in the upstream area (0~7.2 mm), while it is slightly better than that of MCHS in the downstream area (7.2~14.4 mm) because the boundary layers of I-MPFHS cannot reach full development with discontinuous walls. When Re is higher than 550, the flow structure in the back of the micro pin fins turns into an unsteady Kármán vortex street and the heat transfer performance of the micro pin fins is greatly enhanced, especially in the downstream area. Consequently, as shown in Figure 8b, the temperature uniformity of I-MPFHS at Re = 678.7 is much better than that of MCHS.

Besides, it is found that the rising trends of the temperature curves of MCHS and I-MPFHS in Figure 8 stop at the last points. The reason for this phenomenon is that the last temperature sensor is very close to the border of the heated area. At the border of the heated area, a part of the input heating power is conducted to the unheated area, resulting in an apparent temperature reduction. For S-MPFHS, the heat transfer performance of the convective surfaces is high so that the lateral conduction is not severe. As a result, the temperature reduction near the last temperature sensor in S-MPFHS is much lower than those in MCHS and I-MPFHS.

The pressure drop between the inlet and the outlet is another important parameter in evaluating the performance of a heat sink. Figure 9 shows variations of the pressure drop with Re. It shows that the pressure drop of S-MPFHS is much higher than that of MCHS and I-MPFHS because of the tortuous flow path, which not only increases friction resistance by interrupting the development of the boundary layers but also brings on high pressure resistance by generating flow separation in the back of the micro pin fins. As for I-MPFHS, the pressure resistance takes a low proportion of the total resistance and the pressure drop of the heat sink is dominated by friction resistance at low Re, so that the pressure drop characteristic of I-MPFHS is similar to that of MCHS at low Re. However, when Re increases, the separation zones in the back of the micro pin fins gradually become unsteady. As a result, the proportion of pressure resistance increases and the pressure drop difference between I-MPFHS and MCHS becomes larger.

The analysis above shows that a strong cooling capability is usually accompanied by a high pressure drop and so it is difficult to choose the best heat sink simply via Figure 7 and Figure 9. Therefore, it is necessary to compare the average thermal resistances of MCHS, I-MPFHS and S-MPFHS under conditions of the same power consumption or the same pressure drop to evaluate the comprehensive performance of these three heat sinks.

Figure 10 shows the variations of average thermal resistance (*θ*) with the power consumption of the water pump. The power consumption is calculated by:(10)Power consumption = ΔP×Q.

These results show that for low power consumption, the endwall effect strongly suppresses the heat transfer performance of micro pin fins. Thus, S-MPFHS has a small advantage over MCHS and I-MPFHS. For moderate power consumption, the endwall effect diminishes and the vortex shedding of I-MPFHS has not started. In this situation, S-MPFHS shows a much lower *θ* than MCHS and I-MPFHS. However, when the applied pumping power is further increased, the vortex shedding of I-MPFHS starts and the performance advantage of S-MPFHS over I-MPFHS shows a declining trend. In summary, S-MPFHS is constantly superior in the whole power consumption range tested in this study. Thus, S-MPFHS is preferred if the pump of the cooling system can provide enough pressure drop, especially under conditions of moderate power consumption.

However, if the pump of the cooling system cannot provide enough pressure drop, the pressure drop will be the main constraint on the cooling system rather than the power consumption. Figure 11 shows variations of the average thermal resistance (*θ*) with the pressure drop. When the pressure drop is lower than 1.5 kPa, the cooling capability of I-MPFHS (*θ* = 0.8513 K/W at ∆P = 1.16 kPa) is a little higher than MCHS (*θ* = 0.8250 K/W at ∆P = 1.18 kPa) and both of them are obviously better than S-MPFHS (*θ* = 0.9613 K/W at ∆P = 1.12 kPa) because of the severe endwall effect and high pressure drop in S-MPFHS. When the pressure drop increases, the endwall effect in S-MPFHS diminishes and the cooling advantage of S-MPFHS becomes evident. However, when the pressure drop is higher than 20 kPa, the cooling capability of I-MPFHS (*θ* = 0.2166 K/W at ∆P = 27.70 kPa) exceeds S-MPFHS (*θ* = 0.2231 K/W at ∆P = 29.49 kPa) because of the high heat transfer enhancement and the low pressure drop price brought by the unsteady vortex street in I-MPFHS. In summary, I-MPFHS shows the most effective cooling capability when the rated pressure drop of the pump is lower than 1.5 kPa or higher than 20 kPa and S-MPFHS has the most effective cooling capability when the rated pressure drop of the pump is in the range of 1.5 to 20 kPa.

## 4. Conclusions

In this study, on-chip cooling systems fabricated by micromachining and silicon-to-silicon bonding are created to study the cooling capabilities of microchannel, inline micro-pin-fin and staggered micro-pin-fin heat sinks (MCHS, I-MPFHS and S-MPFHS) in the Reynolds number range of 79.2 to 882.3. The results show that:

(1)The cooling capabilities and temperature uniformity of S-MPFHS are better than those of MCHS and I-MPFHS at same Re because of the tortuous flow path in S-MPFHS, which improves the synergy between the velocity vector and the temperature gradient and enhances the heat transfer performance. However, at low Re, the endwall effect suppresses the disturbance generated by the staggered micro pin fins and the performance advantage of S-MPFHS decreases.(2)A transition point of Re = 550 is observed for the heat transfer performance of I-MPFHS. For Re < 550, the flow structure in the separation zones of the inline micro pin fins is a steady large-scale vortex pair. The vortex pair rotates at a low speed and it is almost isolated from the main flow, which causes poor heat transfer performance in the separation zones. For Re > 550, the flow stability in the separation zones of the inline micro pin fin decreases and the flow structure in the separation zones gradually evolves from a steady vortex pair to an unsteady Kármán vortex street. Affected by the periodic vortex generation and shedding of the Kármán vortex street, the momentum and energy exchanges between the separation zones and the main flow are significantly enhanced.(3)By comparing the cooling capability of MCHS, I-MPFHS and S-MPFHS for the same power consumption and the same pressure drop of the water pump respectively, we conclude that S-MPFHS is preferred if the pump of the cooling system can provide enough pressure drop. However, for the same pressure drop, I-MPFHS is the best choice when the rated pressure drop of the pump is lower than 1.5 kPa or higher than 20 kPa but S-MPFHS is still preferred when the rated pressure drop of the pump is in the range of 1.5 to 20 kPa.

## Figures and Tables

**Figure 1 sensors-20-05533-f001:**
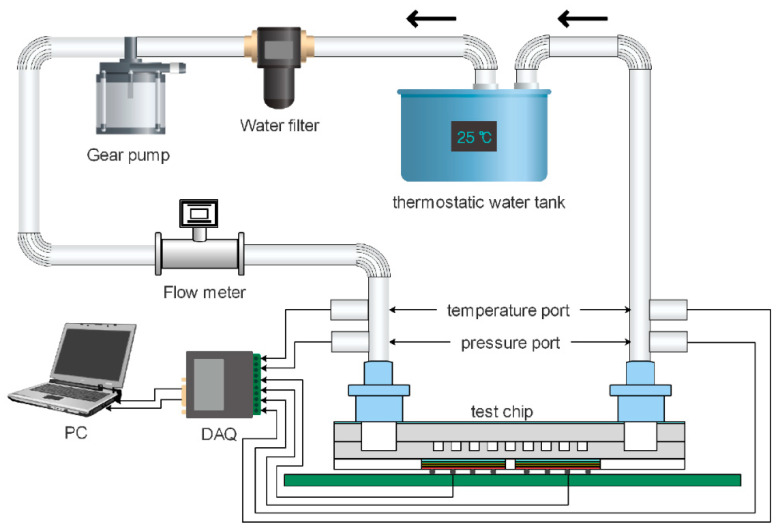
The experimental setup.

**Figure 2 sensors-20-05533-f002:**
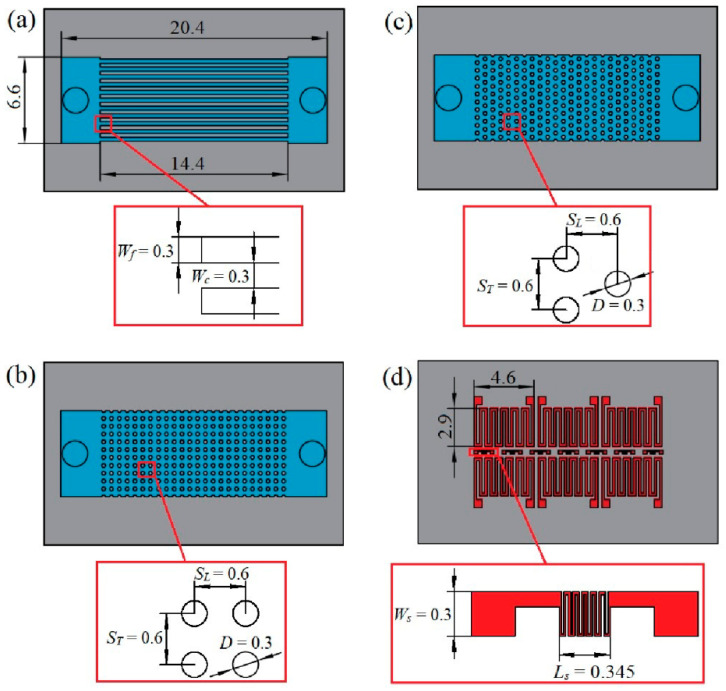
Structural illustration of the test chips: (**a**) the microchannel heat sink; (**b**) the inline micro-pin-fin heat sink; (**c**) the staggered micro-pin-fin heat sink; (**d**) the thin-film Ti/Pt heaters and sensors. (units: mm).

**Figure 3 sensors-20-05533-f003:**
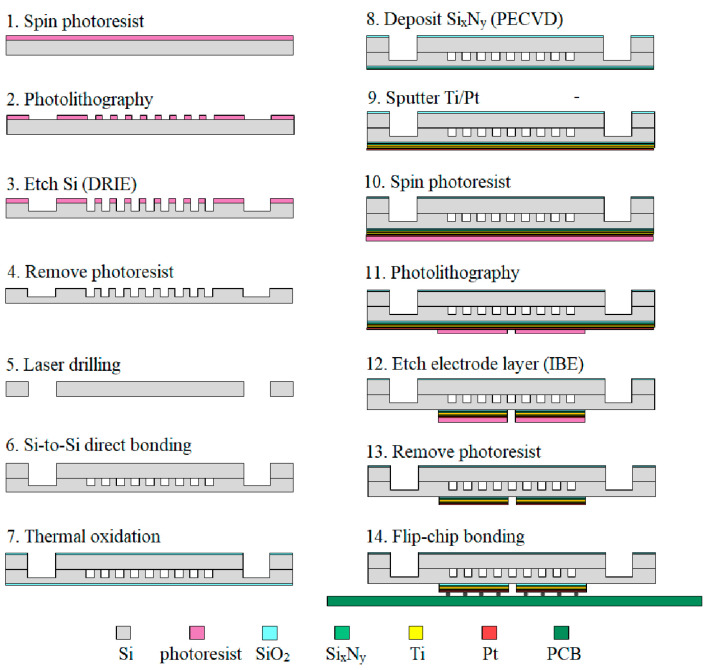
Fabrication process of the test chips.

**Figure 4 sensors-20-05533-f004:**
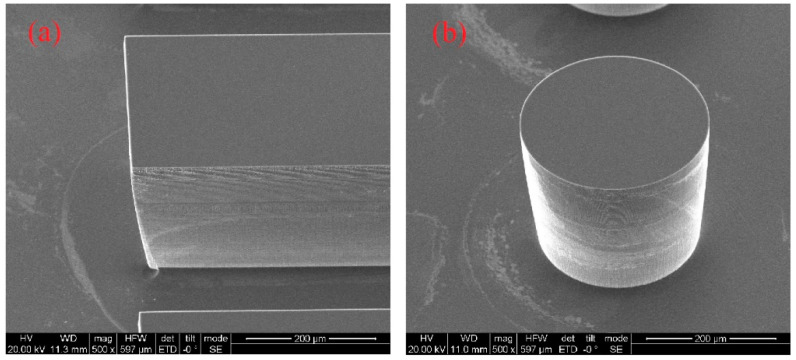
Scanning electron microscopy (SEM) photos of (**a**) the microchannel and (**b**) the micro pin fin.

**Figure 5 sensors-20-05533-f005:**
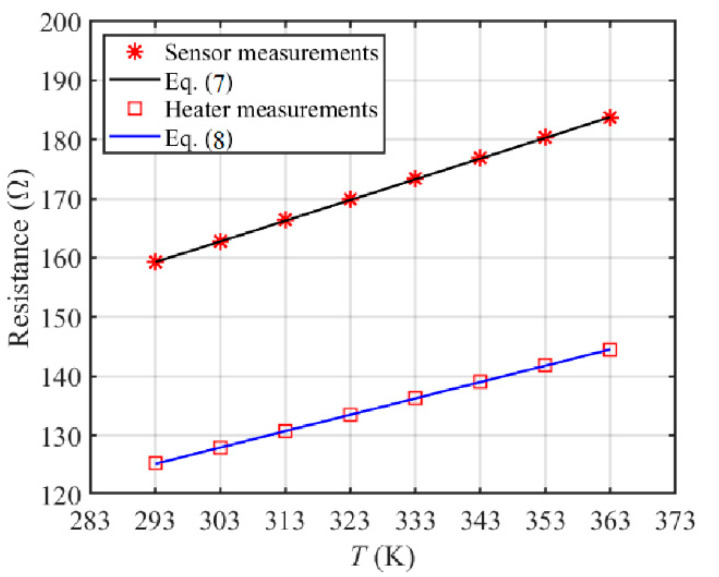
Calibration curves for thin-film Ti/Pt sensors and heaters.

**Figure 6 sensors-20-05533-f006:**
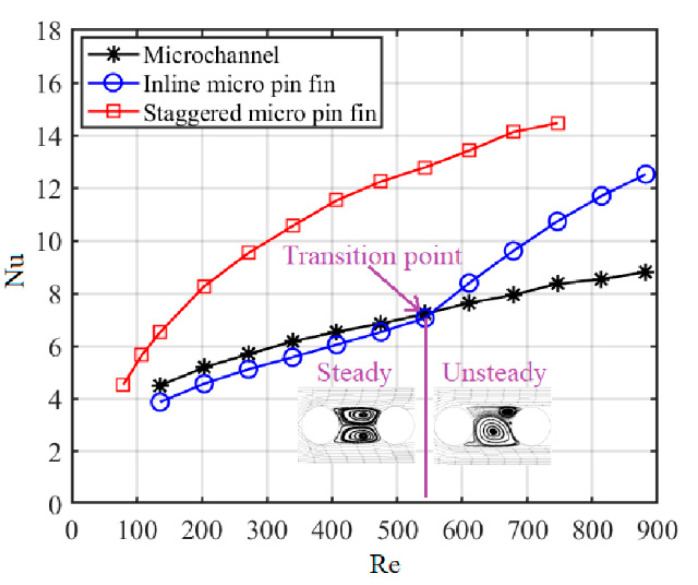
Variations of the average Nusselt number with the Reynolds number.

**Figure 7 sensors-20-05533-f007:**
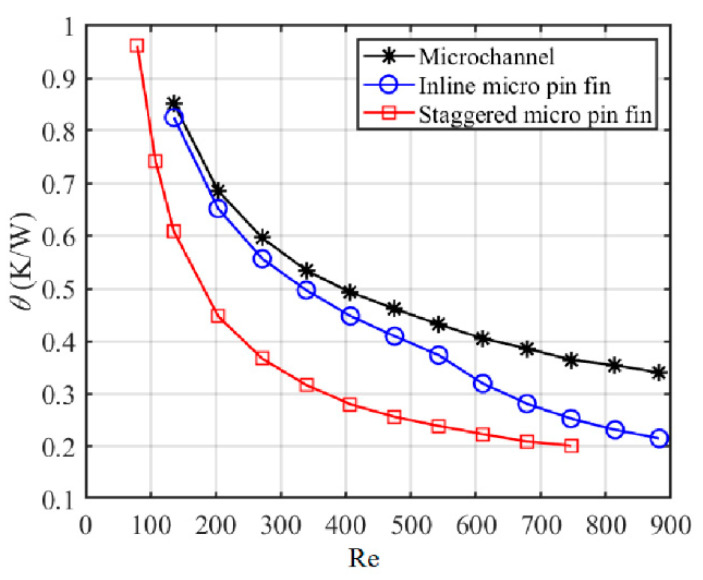
Variations of the average thermal resistance with the Reynolds number.

**Figure 8 sensors-20-05533-f008:**
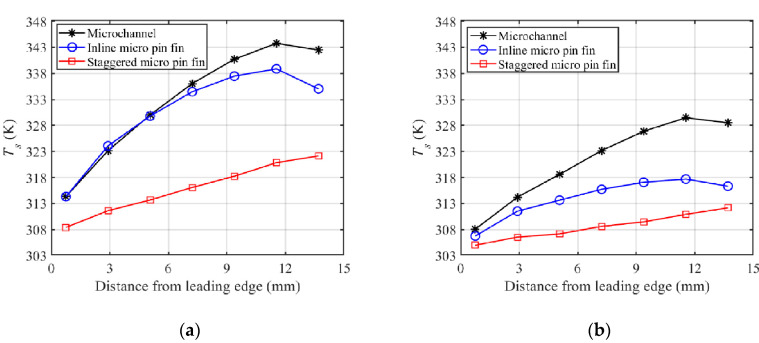
Temperature distributions along the centerline of the simulated power IC at (**a**) Re = 339.3, (**b**) Re = 678.7.

**Figure 9 sensors-20-05533-f009:**
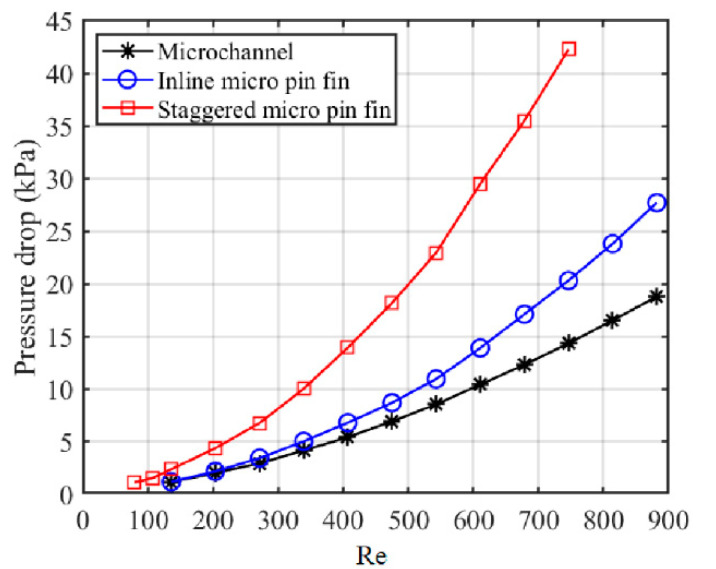
Variations of the pressure drop with the Reynolds number.

**Figure 10 sensors-20-05533-f010:**
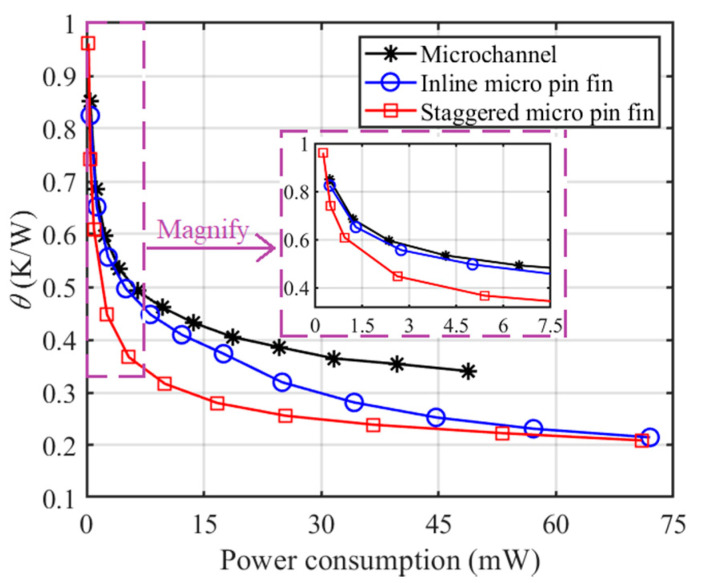
Variations of the average thermal resistance with power consumption.

**Figure 11 sensors-20-05533-f011:**
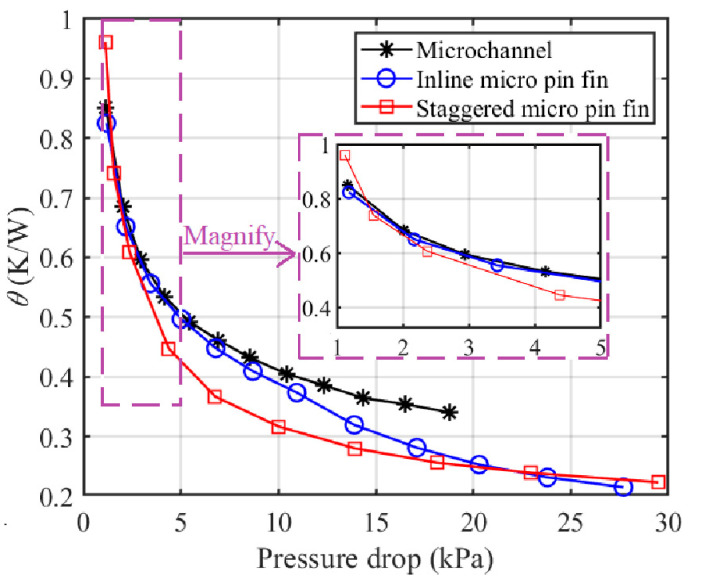
Variations of the average thermal resistance with the pressure drop.

**Table 1 sensors-20-05533-t001:** Experimental Uncertainties.

	Parameters	Uncertainty
**Measurements**	Flow rate, *Q*	±0.5%
	Geometric error	±1 μm
	Inlet water temperature, *T_i_*	±0.2 K
	Outlet fluid temperature, *T_o_*	±0.2 K
	Temperature of sensors, *T_s_*	±0.42 K
	Temperature of heaters, *T_h_*	±0.45 K
	Pressure drop, ∆P	±4.5% (maximum)
**Derived parameters**	Reynolds number, Re	±0.8% (maximum)
	Nusselt number, Nu	±5.7% (maximum)
	Thermal resistance, *θ*	±5.0% (maximum)

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
