# Peer review of "An Experimental Study of Microchannel and Micro-Pin-Fin Based On-Chip Cooling Systems with Silicon-to-Silicon Direct Bonding"

_sensors, 2020, doi:10.3390/s20195533_

Round 1

Reviewer 1 Report

The scope of the paper is not appropriate to the journal aim. Authors should submit to a more appropriate journal

Author Response

The scope of the paper is not appropriate to the journal aim. Authors should submit to a more appropriate journal.

Answer: We rechecked the scope of the journal Sensors, and found one related topic: MEMS/NEMS. As we know, the MEMS based thermal management technology is a promising technology for high-performance microchips. Moreover, two similar researches have been published on the journal Sensors:

[1] Wang T.; Wang J.J.; He J.; et al. A Comprehensive Study of a Micro-Channel Heat Sink using Integrated Thin-Film Temperature Sensors. Sensors 2018, 18, 299.

[2] Wang T.; Wang J.J; He J.; et al. Investigation of the Temperature Fluctuation of Single-Phase Fluid Based Microchannel Heat Sink. Sensors, 2018, 18, 1498.

Therefore, we plead the reviewer to reconsider the decision on our manuscript.

Reviewer 2 Report

The authors described the single-phase heat transfer in microchannel and micro-pin-fin cooling system. 

The cooling system was installed by Si-Si direct bonding, which is practically important.

I consider the overall merit of this paper is enough to publish, however, I would like to clarify some parts before acceptance.

  • How did you design the microchannel heat sink? Is there any equivalent parameter with micro-pin-fin? (ex. total surface area, or frictional coefficient) It is difficult to compare between microchannel and micro-pin-fin without any consideration.
  • How did you obtain the streamline in Figure 6? Are these images from simulation, experiment, or schematic? 

Author Response

â‘  How did you design the microchannel heat sink? Is there any equivalent parameter with micro-pin-fin? (ex. total surface area, or frictional coefficient) It is difficult to compare between microchannel and micro-pin-fin without any consideration.

Answer: The equivalent parameter between the microchannel heat sink and the micro-pin-fin heat sink is hydraulic diameter, which is the basis to calculate Reynolds number, Nusselt number, and frictional coefficient. Actually, the inline micro-pin-fin heat sink is an optimized structure for the microchannel heat sink. As we know, the boundary layer develops with the flow distance in a microchannel, resulting a thick thermal boundary layer and a low heat transfer performance in the downstream of the microchannel. So, inspired by the concept of redeveloping boundary layer, the microchannel heat sink is optimized to the inline micro-pin-fin heat sink with discontinuous side walls to interrupt the development of boundary layer. However, the heat transfer performance in the separation zones of the inline micro pin fins is low, so a new heat sink structure with staggered micro pin fins is proposed to decrease the size of the separation zones.

â‘¡ How did you obtain the streamline in Figure 6? Are these images from simulation, experiment, or schematic?

Answer: The streamline images in Fig. 6 are calculated by 2D numerical simulations and used as schematics to help explain the change of the Nu-Re curve.

Reviewer 3 Report

In this work, an experimental study is carried out in order to compare dynamic and thermal performances of three different water cooled micro heat sinks fabricated through MEMS micromachining and silicon-to-silicon bonding. The paper is, in my opinion, well organized and written. I suggest minor revisions. Here are some notes, in order of line/page:

* Reynolds number ranges from 79.2 to 882.3, why have these values been chosen? What water inlet velocities do these values correspond to?

* I would remove the only equation in introduction (equation 1, TPI definition), the TPI index is not even used in the paper, the index can just be cited or else described by words it in the text.

* Figure 2c, please add symbol “D” for pin diameter in the sketch.

* Line 142, temperatures are referred to the fluid, I would add in the sentence “where Ti and To are the average temperatures of THE FLUID AT THE chip inlet and outlet, respectively”

* Line 179-187: although the onset of unsteady Kármán vortices is well known in such geometries, it cannot be directly seen from authors’ experimental results (no flow field data available) but just inferred from the Nusselt Number behavior. Maybe authors should add a reference regarding this phenomenon.

* Concerning figure 8, authors should comment on the change in behaviour of the last points (except for the staggered case), i.e. the temperature decrease at the outlet. Is it an effect of conduction in the silicon upper and lower layers?

* Line 295: I would write “However, for the same pressure drop, …” instead of “if the pump cannot provide enough pressure drop”, because in the same sentence reference is made to a pressure drop higher than 20 kPa.

* Finally, some small style notes: Re and Nu are usually not written in italics; line 32: cm2 instead of cm^2; line 37: what does the acronym TIM stand for? (thermal interface material?), please ad it in the text; line 61: is it TORTUOS?; line 74: I would rather write “four-sided HEATED”.

Author Response

â‘  Reynolds number ranges from 79.2 to 882.3, why have these values been chosen? What water inlet velocities do these values correspond to?

Answer: The lower limit of the tested Reynolds number range for the microchannel heat sink (MCHS) and the inline micro-pin-fin heat sink (I-MPFHS) is 135.7. When Re is lower than 135.7, the water in MCHS and I-MPFHS start to boil. The lower limit of the tested Reynolds number range for the staggered micro-pin-fin heat sink (S-MPFHS) is 79.2. The pressure drop of S-MPFHS at Re = 79.2 is close to that of MC and I-MPFHS at Re = 135.7, which provides convenience in comparing the performance of MCHS, I-MPFHS, and S-MPFHS under the constraint of same pressure drop, as shown in Fig. 11. The upper limit of the tested Reynolds number range 882.3 is chosen for a safe pressure drop of the pump (<50 kPa) and proper uncertainties of the Nusselt number and the thermal resistance. The corresponding inlet velocity range is 0.24 ~ 2.63 m/s.

â‘¡ I would remove the only equation in introduction (equation 1, TPI definition), the TPI index is not even used in the paper, the index can just be cited or else described by words it in the text.

Answer: Thanks for the reviewer’s suggestion. The definition of TPI is removed.

â‘¢ Figure 2c, please add symbol “D” for pin diameter in the sketch.

Answer: Symbol “D” for pin diameter is added in Fig. 2c.

â‘£ Line 142, temperatures are referred to the fluid, I would add in the sentence “where Ti and To are the average temperatures of THE FLUID AT THE chip inlet and outlet, respectively”

Answer: The descriptions of Ti and To are revised according to the reviewer’s suggestion.

⑤ Line 179-187: although the onset of unsteady Kármán vortices is well known in such geometries, it cannot be directly seen from authors’ experimental results (no flow field data available) but just inferred from the Nusselt Number behavior. Maybe authors should add a reference regarding this phenomenon.

Answer: Two references regarding the transition phenomenon in the separation zone of a micro pin fin are added to the manuscript:

[1] Jung J.; Kuo C.J.; Peles Y.; Amitay M. The Flow Field around a Micropillar Confined in a Microchannel. Int. J. Heat Fluid Flow. 2012, 36, 118-132.

[2] Wang Y.Y.; Houshmand F.; Elcock D; Peles Y. Convective Heat Transfer and Mixing Enhancement in a Microchannel with a Pillar. Int. J. Heat Mass Transf. 2013, 62, 553-561.

â‘¥ Concerning figure 8, authors should comment on the change in behavior of the last points (except for the staggered case), i.e. the temperature decrease at the outlet. Is it an effect of conduction in the silicon upper and lower layers?

Answer: Thanks for the reviewer’s suggestion. The explanation of the changes in behavior of the last points in Fig. 8 is added to the manuscript from line 229 to line 235. The last temperature sensor is very close to the border of the heated area. At the border of the heated area, a part of the input heating power is conducted to the unheated area, resulting in an apparent temperature reduction. For staggered micro-pin-fin heat sink, the heat transfer performance of the convective surfaces is high so that the conduction is not severe. As a result, the temperature reduction near the last temperature sensor in S-MPFHS is much lower than those in MCHS and I-MPFHS.

⑦ Line 295: I would write “However, for the same pressure drop, …” instead of “if the pump cannot provide enough pressure drop”, because in the same sentence reference is made to a pressure drop higher than 20 kPa.

Answer: The sentence is revised to “However, for the same pressure drop, I-MPFHS…” according to the reviewer’s suggestion.

â‘§ Finally, some small style notes: Re and Nu are usually not written in italics; line 32: cm2 instead of cm^2; line 37: what does the acronym TIM stand for? (thermal interface material?), please ad it in the text; line 61: is it TORTUOS?; line 74: I would rather write “four-sided HEATED”.

Answer: The mistakes listed above are revised accordingly. We really appreciate the reviewer’s careful work.

Round 2

Reviewer 1 Report

Dear authors, I am convinced by the answer provided. The paper is well written and I don't have further things to address as you have already addresses many issues in the paper highlighted in yellow already.